# The Preventive Role of Physical Activity in Systemic Sclerosis: A Cross-Sectional Study on the Correlation with Clinical Parameters and Disease Progression

**DOI:** 10.3390/ijerph191610303

**Published:** 2022-08-18

**Authors:** Cristina Antinozzi, Elisa Grazioli, Maria De Santis, Francesca Motta, Paolo Sgrò, Federico Mari, Caterina Mauri, Attilio Parisi, Daniela Caporossi, Guglielmo Duranti, Roberta Ceci, Luigi Di Luigi, Ivan Dimauro

**Affiliations:** 1Unit of Endocrinology, Department of Movement, Human and Health Sciences, University of Rome “Foro Italico”, 00135 Rome, Italy; 2Unit of Physical Exercise and Sport Sciences, Department of Movement, Human and Health Sciences, University of Rome “Foro Italico”, 00135 Rome, Italy; 3Department of Experimental and Clinical Medicine, “Magna Graecia” University, 88100 Catanzaro, Italy; 4IRCCS Humanitas Research Hospital—Division of Rheumatology and Clinical Immunology, Via Manzoni 56, Rozzano, 20089 Milan, Italy; 5Department of Biomedical Sciences, Humanitas University, Via Rita Levi Montalcini 4, Pieve Emanuele, 20090 Milan, Italy; 6Unit of Bioengineering and Neuromechanics of Movement, Department of Movement, Human and Health Sciences, University of Rome “Foro Italico”, 00135 Rome, Italy; 7Unit of Biology and Human Genetic, Department of Movement, Human and Health Sciences, University of Rome “Foro Italico”, 00135 Rome, Italy; 8Unit of Biochemistry of Movement, Department of Movement, Human and Health Sciences, University of Rome “Foro Italico”, 00135 Rome, Italy

**Keywords:** physical activity, systemic sclerosis, quality of life, oxidative stress, inflammation, prevention

## Abstract

Although exercise is associated with improved health in many medical conditions, little is known about the possible influences of physical activity (PA) habits pre- and post- a diagnosis of systemic sclerosis (SSc) on disease activity and progression. This cross-sectional study assessed, for the first time, self-reported pre- and post-diagnostic PA levels with the aim to verify if changes in these levels were correlated with demographic/anthropometric data (e.g., weight, height, gender, age, BMI), disease duration, diagnostic/clinical parameters (e.g., skin involvement, pulmonary hemodynamic/echocardiographic data, disease activity) related to disease activity and progression, and quality of life in a population-based sample of patients with SSc. Adult participants (*n* = 34, age 56.6 ± 13.3 years) with SSc (limited cutaneous SSc, lcSSc, *n* = 20; diffuse cutaneous SSc, dcSSc, *n* = 9; sine scleroderma SSc, *n* = 5) were enrolled at the Division of Rheumatology and Clinical Immunology of the Humanitas Research Hospital. All medical data were recorded during periodic clinical visits by a rheumatologist. Moreover, all subjects included in this study completed extensive questionnaires to evaluate their health-related quality of life (HRQOL), and others related to health-related physical activity performed before (PRE) and after (POST) the diagnosis of disease. The linear regression analysis has shown that either a high Sport_index or Leisure_index in the PRE-diagnostic period was correlated with lower disease duration in dcSSc patients. Physical load during sport activity and leisure time accounted for ~61.1% and ~52.6% of the individual variation in disease duration, respectively. In lcSSc patients, a high PRE value related to physical load during sporting activities was correlated with a low pulmonary artery systolic pressure (sPAP) and the POST value of the Work_index was positively correlated with the left ventricular ejection fraction (LVEF), and negatively with creatine kinase levels (CK). Interestingly, the univariate analysis showed that Work_index accounts for ~29.4% of the variance in LVEF. Our analysis clearly reinforces the concept that high levels of physical load may play a role in primary prevention—delaying the onset of the disease in those subjects with a family history of SSc—as well as in secondary prevention, improving SSc management through a positive impact on different clinical parameters of the disease. However, it remains a priority to identify a customized physical load in order to minimize the possible negative effects of PA.

## 1. Introduction

Systemic sclerosis (SSc) is a rare, systemic autoimmune disease characterized by skin fibrosis and vasculopathy [1]. Multiple systems (e.g., musculoskeletal, cardiovascular, pulmonary, and gastrointestinal) are involved, resulting in a broad range of symptoms [2]. Depending on the skin’s involvement, one can distinguish between limited cutaneous SSc (lcSSc), which manifests with only partial skin and minor systemic involvement; diffuse cutaneous SSc (dcSSc), which includes extensive skin and systemic involvement; and SSc *sine*, with no evident skin involvement [3].

The prevalence of SSc ranges from 3.1 to 144.5 per 100,000 person-years, with a pooled prevalence of 17.6 (95% CI, 15.1–20.5) per 100,000 person-years (*I*^2^ = 100%)—mainly affecting women in the prime of their life [4,5].

Generally, SSc is associated with significant morbidity, including pain, disability, depression, and reduced quality of life [6,7,8,9]. Moreover, reduced physical activity (PA) and physical capacity in performing daily life activities has been also observed [10,11,12]. In fact, SSc may lead to muscle weakening [13] and impairment in oxygen transport and consumption [14], contributing significantly to reduced physical performances and poor health-related quality of life (HRQL) [15].

Regular PA produces great benefits in healthy and unhealthy individuals of different genders and ages. In particular, it improves muscular and functional performance and endocrine–metabolic, cardiovascular and mental health, as well as having a systemic anti-aging effect through different mechanisms including the modulation of redox homeostasis and/or stress-response proteins [16,17,18,19,20,21,22,23,24,25,26,27,28,29].

Specifically, in rheumatic diseases, regular PA is fundamental in rehabilitation processes. It improves the overall functioning of the individual and supports patients in meeting the demands of daily living [30,31,32,33]. Although the scientific evidence on the effects and safety of PA in SSc is scanty, its positive effects on exercise tolerance, cardiorespiratory fitness, walking distance, and muscle strength and function—as well as HRQL in SSc—are described [34,35,36,37,38,39]. Unfortunately, the existing studies considered fail to show robust evidence. Most of them are also related to small-sized group studies correlating the timing of starting regular PA (i.e., before or after diagnosis) with the common parameters used in clinical settings to predict the course of SSc. Furthermore, it is not clear whether increasing PA per se (i.e., starting either before or after SSc diagnosis) is beneficial for these parameters [35,38,39].

A better understanding of the clinical parameters associated with PA levels would be helpful to identify a customized physical load in order to minimize the possible negative effects of PA, maximizing its effects and thus delaying/countering the onset and progression of the disease. Thus, for the first time, the present cross-sectional exploratory study aims to investigate: (i) self-reported pre- and post-diagnostic PA levels between subtypes of SSc patients—lcSSc and dcSSc—and in general in an all population-based sample of patients with SSc; (ii) if changes in these levels are correlated/associated with demographic data (e.g., weight, height, gender, age, disease duration), diagnostic/clinical parameters (e.g., pulmonary hemodynamic/echocardiographic data, disease activity) related to disease progression, and quality of life measurements.

## 2. Materials and Methods

### 2.1. Study Design and Participants

Volunteers affected by SSc were recruited at the Division of Rheumatology and Clinical Immunology of the Humanitas Research Hospital in Milan, Italy, from September 2020 through September 2021. The inclusion criteria were: a diagnosis of SSc, age ≥ 18 years, and having the ability to give informed consent and be able to respond to questionnaires; the exclusion criteria were: genetic diseases affecting locomotion, pulmonary diseases of any other cause, and heart failure. All volunteers were diagnosed for SSc subtypes (lcSSc, *n* = 20; dcSSc, *n* = 9; sine scleroderma, *n* = 5) according to the European Scleroderma Trial and Research (EUSTAR) recommendations [3,40] and were positive for anti-Scl-70 and antocentromere antibodies (ACA).

All subjects were asked to give written informed consent to participate in a cross-sectional exploratory study, which was approved by the University Committee for Research (CAR), University of Rome Foro Italico (CAR91/2021).

All medically sensitive data were analyzed anonymously. Demographic/anthropometric and diagnostic/clinical parameters (Table 1 and Table 2), known to be valid indicators of disease progression, were investigated as described previously [41,42,43,44] and recorded during periodic clinical visits by a rheumatologist. All volunteers recruited in this study also completed extensive questionnaires relating to the individual’s perception of their position in life in the context of the culture and value systems in which they live and about their goals, expectations, standards, and concerns (also known as health-related quality of life (HRQOL), through the Short Form Health Survey, SF-36) [45,46], as well as to health-related physical activity performed before (PRE) and after (POST) their diagnosis with the disease through the modified Baecke Questionnaire for Older Adults [47].

### 2.2. Quality of Life Assessment (SF-36)

For the purpose of assessing their current quality of life, all participants were asked to complete the Short-Form Health Survey questionnaire (SF-36). It is commonly used as a generic measure to assess HRQOL. It has gained popularity as a means of evaluating outcomes in a wide variety of patient groups and surveys. The SF-36 questionnaire consists of 36 items, which are used to calculate eight subscales: physical functioning (PF), role physical (RP), bodily pain (BP), general health (GH), vitality (VT), social functioning (SF), role emotional (RE), and mental health (MH). The first four scores can be summed to create the physical composite score (PCS), while the last four can be summed to create the mental composite score (MCS). Scores for the SF-36 scales range between 0 and 100, with higher scores indicating a better HRQOL [45,46].

### 2.3. Assessment of Habitual Physical Activity

All participants were asked to complete questionnaires relative to health-related physical activity performed before (PRE) and after (POST) the diagnosis at one time. In this study, the Modified Baecke Questionnaire for Older Adults for physical activity was used to determine the level of PA. This tool is used to estimate the level of PA based on the patient’s self-report. The questionnaire was composed of 16 items testing the physical load at work, physical load during sport activities, and physical load during leisure time. Three basic indices resulted from the application of this questionnaire: the Work Index, Sport Index, and Leisure Index. The lowest possible value of the indices was 1.0 unit—signifying the lowest physical activity—while the highest possible value was 5.0 units, signifying the highest PA [47]. The original version of this questionnaire was validated in both healthy and diseased populations [48,49].

### 2.4. Statistical Analyses

We envisioned this as an explorative study; no sample size calculation was performed. All statistical analyses were performed using the Statistical Package for the Social Sciences, version 22.0 (IBM SPSS Statistics, Milan, Italy) and GraphPad Prism software 8.0 (GraphPad Software, San Diego, CA, USA).

For statistical analysis, patients with SSc who fell into the sine subtype, with symptoms and signs similar to patients with lcSSc, were included in this group.

Quantitative variables were expressed as mean (±SD) and qualitative variables as proportions. Normality tests were run to check whether data were normally distributed. Distributions of non-normally distributed variables were also studied. Unpaired/paired *t*-tests/Mann–Whitney U tests were used to compare quantitative variables and the Chi-square test or Fischer’s exact were used to compare qualitative variables.

A correlation matrix was used to test the strength of any associations between variables. To analyze the relationship between all variables and the contribution of each independent variable, a single/multiple linear regression analysis with forward selection or/and backward elimination of variables was assessed. To avoid multicollinearity, only one variable was selected and entered into the model in case of a significant correlation (r > 0.7) between two variables. Statistical significance was defined as *p* < 0.05.

## 3. Results

### 3.1. Characteristics of the Study Population

The demographic characteristics of patients are summarized in Table 1. In particular, the sample consisted of 34 patients (94.1% women) with a mean age of 56.6 ± 13.3 years. The average time since the onset of non-Raynaud’s symptoms was 85.1 ± 64.8 months and over half of the patients were diagnosed with limited or sine SSc (*n* = 25, 73.5%). No significant differences were observed in any demographic parameters between the two SSc subtypes (lcSSc vs. dcSSc, *p* > 0.05; Table 1).

### 3.2. Clinical Parameters

Almost all volunteers had an estimated systolic pulmonary artery pressure (sPAP) ≤ 36 mmHg, and only one had pulmonary arterial hypertension (PAH).

As expected, the modified Rodnan skin score (mRSS), a surrogate measure of disease severity and mortality in SSc patients, was significantly higher in dcSSc compared with lcSSc (lcSSc vs. dcSSc, 3.6 ± 3.1 vs. 10.9 ± 5.9, *p* = 0.0008). Similarly, dcSSc showed a highest percentage of myocarditis (lcSSc vs. dcSSc, 4.0% vs. 44.4%, *p* = 0.003), interstitial lung disease (ILD; lcSSc vs. dcSSc, 24.0 vs. 66.7, *p* = 0.021), and a significant reduction in the forced vital capacity (FVC) of the percent predicted (%pred; lcSSc vs. dcSSc, 105.4 ± 19.8 vs. 86.6 ± 10.1, *p* = 0.012).

Finally, in dcSSc patients only, there was a significant sign of disease activity (lcSSc vs. dcSSc, 20% vs. 66.7%, *p* = 0.010).

No significant differences were observed in all other clinical parameters between both SSc subtypes (lcSSc vs. dcSSc, *p* > 0.05; Table 2).

### 3.3. Quality of Life Measurement

The general evaluation of the quality of life revealed a significant difference only between lcSSc and dcSSc in the mean SF-36 dimension score for bodily pain (lcSSc vs. dcSSc, 70.5 ± 20.7 vs. 52.2 ± 16.6, *p* = 0.026; Table 3).

No significant differences were observed between lcSSc and dcSSc for all other domain scores (*p* > 0.05; Table 3).

### 3.4. Physical Activity before and after Diagnosis of SSc

As shown in Table 4, the analysis of individuals’ habitual physical activity before (PRE) and after (POST) SSc diagnosis highlighted different aspects depending upon the consideration of all subjects as a whole or taking into account the exact subtype of SSc for each patient. In particular, we found that the whole SSc group showed a significant reduction in both work (PRE vs. POST, 2.6 ± 0.6 vs. 2.3 ± 1.0, *p* < 0.043) and leisure (PRE vs. POST, 2.8 ± 0.8 vs. 2.4 ± 1.1, *p* < 0.009) indices, whereas no change was observed for the sport index (*p* > 0.05). When this analysis was performed at the SSc subtypes level, there were significant differences between the work (PRE vs. POST, 2.5 ± 0.6 vs. 2.1 ± 1.1, *p* < 0.048) and leisure (PRE vs. POST, 2.8 ± 0.9 vs. 2.3 ± 1.2, *p* < 0.018) indices of lcSSc. No other significant change was observed between or within SSc subtype groups (*p* > 0.05).

### 3.5. Correlations and Regression Analysis

To evaluate associations among PA levels, before (PRE) and after (POST) SSc diagnosis, and clinical parameters—as well as to describe how they are dependent numerically on all indices of physical activity levels (i.e., work, sport and, leisure)—a correlation matrix and a regression analysis were performed, respectively.

In general, without distinction between SSc subtypes, the correlation analysis showed that the PRE value of Work_index correlated positively with the diffusion capacity of the lung for carbon monoxide (DLCO; r = 0.498, *p* = 0.003) and C-reactive protein (CRP; r = 0.400, *p* = 0.019), whereas the POST value correlated with the left ventricular ejection fraction (LVEF; r = 0.349, *p* = 0.034) and mRSS (r = 0.439, *p* = 0.005; Figure 1A,B).

Although both the PRE and POST value of Sport_index showed no significant correlations with clinical parameters (*p* > 0.05), it was interesting to note a negative correlation of POST values with body mass index (BMI; r = −0.356, *p* = 0.023) and weight (r = −0.308, *p* = 0.040), as well as Role_limit PH (r = −0.358, *p* = 0.025) and Role_limit EP (r = 0.498, *p* = 0.003). Similarly, the PRE value of the leisure index was negatively correlated with weight (r = −0.353, *p* = 0.022), whereas the POST value had a negative correlation with BMI (r = −0.358, *p* = 0.020) and weight (r = −0.353, *p* = 0.022; Figure 1A,B).

In dcSSc, the correlation analysis highlighted only a negative correlation of disease duration with the PRE values of Sport (r = −0.777, *p* = 0.019) and Leisure (r = −0.751, *p* = 0.023) _index (Figure 1C,D). No significant correlations were observed with consideration to physical activity indices POST-diagnosis and all other parameters (*p* > 0.05).

Interestingly, in lcSSc, there were both positive correlations among physical load at work post-diagnosis with LVEF (r = 0.513, *p* = 0.012) and mRSS (r = 0.648, *p* = 0.000), but also a negative correlation with CK (r = −0.585, *p* = 0.019; Figure 1E,F). Furthermore, negative correlations were found either between physical load during sport activity at PRE with health change (r = −0.366, *p* = 0.047) and sPAP (r = −0.450, *p* = 0.018), or at POST with weight (r = −0.379, *p* = 0.034)—whereas positive correlations were found only between he PRE values of sport index and pain (r = 0.381, *p* = 0.040; Figure 1E,F).

Finally, the leisure_index at PRE was positively correlated with C-reactive protein (CRP; r = 0.386, *p* = 0.038), and at POST was negatively correlated with weight (r = −0.373, *p* = 0.036).

The univariate analysis identified both PRE values of Sport_index and Leisure_index as significant independent variables associated with disease duration, each accounting for ≈61.1% (F = 11, *p* = 0.0128, R square = 0.6112) and ≈ 52.6% (F = 7.769, *p* = 0.027, R square = 0.5260) of interindividual variation in disease duration in dcSSc patients, respectively (Figure 2A,B), whereas their joint contribution to disease duration derived from multivariate analysis resulted not significantly predictive when analysed together (F = 2.901, *p* = 0.141).

As shown in Figure 2C,D, the POST value of Work_index resulted as significant independent variable associated with LVEF and mRSS, accounting for ≈ 29.4% (F = 7.061, *p* = 0.017, R square = 0.2935) and ≈ 26.3% (F = 7.862, *p* = 0.010, R square = 0.2633), respectively (Figure 2C,D).

All others univariate analysis, performed based on correlation data, resulted not predictive of selected dependent variables (*p* > 0.05).

Figure 3 highlights the asymmetric distribution of the non-normal quantitative variables, which should be read together with the (mean, standard deviation) pairs in Table 2. It is worth noticing the common pattern among such variables, showing rather high variance and a strong concentration around their median.

## 4. Discussion

To our knowledge, this explorative cross-sectional study is the first to evaluate in SSc patients—as a whole or divided into scleroderma subtypes (i.e., lcSSc and dcSSc)—the possible relationships between the subjective measures of habitual PA and diagnostic/clinical parameters related to disease progression and quality of life measurements.

In general, we found a significant reduction in PA levels after disease diagnosis. Moreover, PA indices (i.e., Work_, Sport_ and Leisure_index) at the PRE- and POST-diagnostic period showed not only significant associations with several clinical parameters, but some of them have been shown to be significant independent variables associated (predictive) with disease duration, LVEF, and mRSS.

We strongly believe that these results not only confirm the usefulness of regular PA in the primary and secondary prevention of this disease, but that they could give important indications on which to base a longitudinal study capable of evaluating the effectiveness of a customized physical load on selected clinical parameters—indicators of the disease condition.

Physical activity intolerance is an important pathophysiological consequence of SSc. Indeed, the frequent occurrence of vascular damage and pulmonary fibrosis in SSc patients are likely to contribute to ventilatory, cardiocirculatory, or gas-exchange limitations, causing diminished PA capacity.

In this research study, a careful analysis of the level of habitual PA highlighted a general reduction in the physical load at work and during leisure time post-diagnosis with SSc. Interestingly, these changes were significant overall in lcSSc patients. Indeed, although dcSSc patients had clear signs of increased disease activity—including deterioration in quality of life due to increased body pain, as well as a pulmonary and cardiac involvement—in the post-diagnostic period they maintained a comparable physical load in all three areas considered (i.e., work, sport, leisure). Probably, these differences were due to the different number of subjects recruited into the two groups—especially the limited number of dcSSc patients. Indeed, a previous study recruiting a high number of SSc patients (*n* = 752) reported similar levels of PA between patients with limited or sine versus diffuse SSc [50].

In patients with SSc, a physically active lifestyle has already proved to be associated with enhanced tolerance and aerobic capacity [35,51,52], muscle strength [34,52], hand mobility [53,54,55], function in daily activities [54], and health-related quality of life [53,54], even in patients with some degree of lung involvement. In general, we showed interesting correlations among PA indices—referred to as the PRE- and POST-diagnostic period—with several parameters, including DLCO, LVEF, CK, sPAP, BMI, and weight. Specifically, we observed in dcSSc patients how either a high Sport_index or Leisure_index in the PRE-diagnostic period correlated with lower disease duration. Indeed, from the univariate analysis, we found that the physical load during sporting activities and leisure time explained ~61.1% and ~52.6% of the individual variation in disease duration, respectively. In lcSSc patients, a high PRE value related to physical load during sporting activities was correlated with a low sPAP, and POST values of Work_index were positively correlated with LVEF and negatively with CK. Interestingly, the univariate analysis showed that Work_index accounted for ~29.4% of the variance in LVEF.

Correlations between PA and several clinical parameters such as DLCO [56], LVEF [57], and sPAP [58] have been already described; therefore, our results reinforce the general concept that high levels of physical load could result in improvements in numerous clinical parameters. However, we cannot ignore the possibility that an excessive physical load in all three areas examined may have a negative effect on the parameters considered. Indeed, the physical load during work, sport, and leisure activity could have an adequate intensity, volume, and duration to negatively affect specific clinical or HRQL parameters such as those observed in our study, such as pain, CRP, mRSS, and health changes. In these patients, the possible impact that activities of daily living have on immunological and antioxidant defenses—which have received considerable attention since they have been suggested to contribute to the clinical manifestation of SSc—is still controversial [59,60,61,62,63,64]. Additionally, it should be considered that SSc patients have a reduced antioxidant capacity and lower plasma levels of ascorbic acid, α-tocopherol, β-carotene, and selenium than healthy controls [60,65]. For this population, it could be that physical loads during daily activities were relatively high intensity, causing frequent exposure to bursts of systemic inflammation and oxidative stress. At present, no data related to oxidative stress level during daily activities are available, but the analysis of clinical parameters (i.e., mRSS, CK) associated with oxidative stress showed significant correlations [66,67]. Therefore, for this disease condition, we hypothesize that an excessive physical load could contribute to redox homeostasis unbalance, inducing negative effects.

## 5. Limitations and Future Perspectives

The present study has some limitations. First, SSc is a rare disease, and—similar to other studies [39,68,69]—our sample size was small. An increase in the sample size might bring out other significant differences among the parameters observed in this study. Secondly, although the Baecke self-report questionnaire provides useful descriptive information on physical activity levels, it could be interesting to confirm our data by utilizing objective measures—for instance, using accelerometers—to collect information about time per day spent in different PA levels/indices. Moreover, to better understand all the variables observed, future studies should include an age-matched and disease-free control group.

As a future prospective in the field of exercise biology for subjects with specific needs, such as SSc patients, we recommend new scientific research also focusing on the molecular aspects of PA. It is known that PA induces beneficial effects through the stimulation of biological molecules and epigenetic modifications, which participates in numerous biological processes [16,17,19,20,22,23,27,29,70,71,72]. However, all the published studies have identified their primary and secondary outcomes as VO2peak, muscle strength and function, life-satisfaction, Raynaud’s pain, transcutaneous oxygen pressure in finger, and the Six-minute walk test [73]. The molecular analysis of PA for systemic sclerosis would allow the customization of a motor protocol capable, for instance, of stimulating/strengthening redox homeostasis, which is known to be compromised during the establishment and maintenance of disease [59,61,62,74]

## 6. Conclusions

To conclude, our analysis clearly reinforces the concept that regular PA may play a role in primary prevention—delaying the onset of the disease in those with a family history—as well as in secondary prevention, improving SSc management through a positive impact on different clinical parameters of the disease. We strongly believe that a deep knowledge of the associations/correlations between clinical parameters and PA levels, before and after the disease diagnosis period, would be helpful for identifying a customized physical load in order to minimize from one side its possible negative effects, while from the other side delaying/countering the onset and progression of the disease. Moreover, the SSc post-diagnosis health process should include professional counseling to inform patients about the importance of avoiding sedentary behavior and monitoring patients’ PA levels, aerobic capacity, and muscle strength at least once a year, never forgetting that PA should complement pharmacological treatments.

## Figures and Tables

**Figure 1 ijerph-19-10303-f001:**
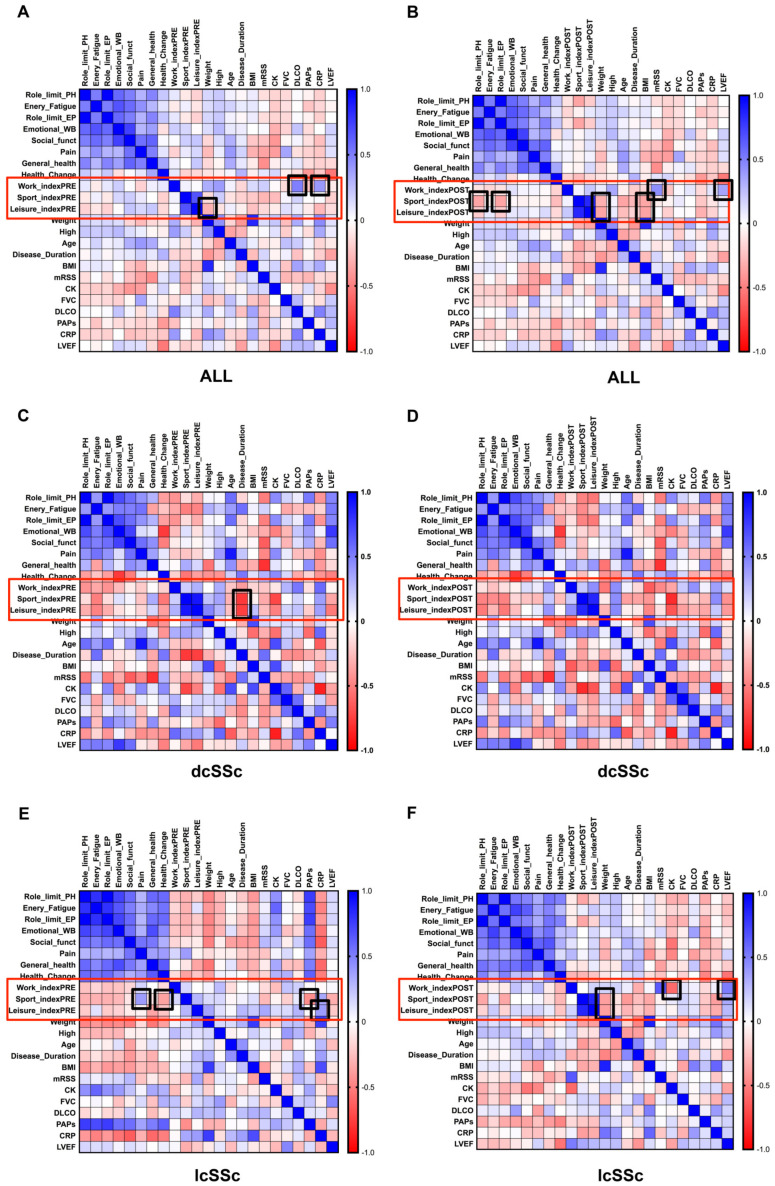
Heat map representation of a correlation matrix of demographic data, diagnostic/clinical parameters related to disease activity and progression, quality of life, and habitual physical activity in ALL SSc (**A**,**B**), dcSSc (**C**,**D**) and lcSSc (**E**,**F**) patients, considering the PRE and POST values of physical activity indices. dcSSc, diffuse cutaneous SSc; lcSSc, limited cutaneous SSc.

**Figure 2 ijerph-19-10303-f002:**
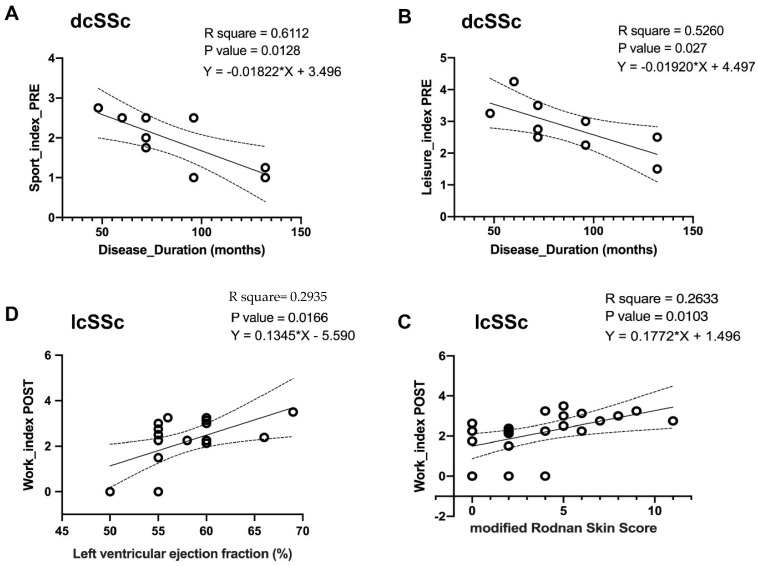
Linear regression analysis between indices for physical activity level and clinical parameters in dcSSc (**A**,**B**) and lcSSc (**C**,**D**) groups. (----------) 95% confidence band. R square represents the contribution of the independent variable to clinical parameter (dependent variable) in the univariate analysis. A low *p*-value (<0.05) indicates a significant relationship between variables. dcSSc, diffuse cutaneous SSc; lcSSc, limited cutaneous SSc.

**Figure 3 ijerph-19-10303-f003:**
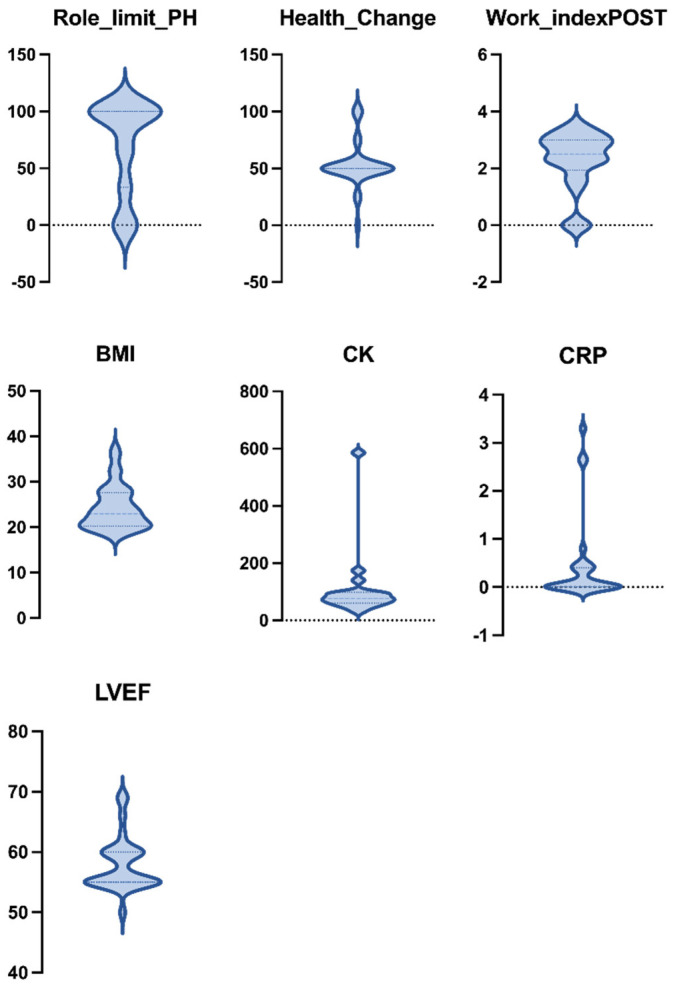
Violin plots representing the distributions of non-normal asymmetric variables over the entire dataset. Dashed and dotted lines represent the median and quartiles, respectively.

**Table 1 ijerph-19-10303-t001:** Demographic characteristics of SSc patients.

Demographic Parameters	Total Sample (*n* = 34)	lcSSc (*n* = 25)	dcSSc (*n* = 9)	*p* Value *
**Age (years)**	56.6 ± 13.3	59.5 ± 13.5	48.8 ± 8.8	>0.05
**Weight (kg)**	65.4 ± 13.6	64.7 ± 13.8	67.2 ± 6.3	>0.05
**Height (cm)**	164.1 ± 7.4	163.1 ± 7.6	166.9 ± 6.3	>0.05
**BMI (kg/m^2^)**	24.3 ± 4.9	24.3 ± 4.8	24.2 ± 5.3	>0.05
**Gender (*n* male/*n* female)**	2/32	2/23	0/9	N/A
**Disease duration (months)**	85.1 ± 64.8	84.5 ± 74.4	86.7 ± 29.9	>0.05

Values are presented as mean ± SD. * *p* < 0.05 lcSSc vs. dcSSc. **N/A**, not applicable; **BMI**, body mass index; **dcSSc**, diffuse cutaneous SSc; **lcSSc**, limited cutaneous SSc.

**Table 2 ijerph-19-10303-t002:** Clinical characteristics of patients.

Clinical Parameters	All Patients (*n* = 34)	lcSSc (*n* = 25)	dcSSc (*n* = 9)	*p* Value *
**CRP (normal: less than 10 mg/L)**	0.49 ± 0.93	0.14 ± 0.21	0.61 ± 1.1	>0.05
**NVC pattern (% cases observed):**				
**Normal**	6.1	4.2	11.1	>0.05
**Early**	21.2	25	11.1	>0.05
**Middle**	21.2	25	11.1	>0.05
**Late**	51.5	45.8	66.7	>0.05
**Disease Activity (% cases observed)**	32.4	20	66.7	**0.010**
**ILD (% cases observed)**	35.3	24.0	66.7	**0.021**
**sPAP (mm Hg)**	27.8 ± 7.0	28.6 ± 8.0	26.2 ± 4.1	>0.05
**FVC (% pred)**	99.7 ± 19.4	105.4 ± 19.8	86.6 ± 10.1	**0.012**
**DLCO (% pred)**	74.6 ± 21.4	79.0 ± 22.0	68.0 ± 24.1	>0.05
**ECG block (% cases observed):**				
**BBDX incomplete**	8.8	12.5	0	>0.05
**Block**	5.9	8.3	0	>0.05
**LVEF%**	57.9 ± 4.5	57.8 ± 4.4	57.0 ± 3.1	>0.05
**Abnormal E/A (% cases observed)**	8.8	12.0	0	>0.05
**mRSS (points)**	5.4 ± 5.0	3.6 ± 3.1	10.9 ± 5.9	**0.0008**
**CK (U/L, normal range 22–198)**	108.0 ± 117.1	112.1 ± 139.1	109.8 ± 46.8	>0.05
**Myocarditis (% cases observed)**	14.7	4.0	44.4	**0.003**

Values of quantitative variables are presented as mean ± SD. Qualitative variables are presented as percentages. * lcSSc vs. dcSSc, Student’s *t* test/Mann–Whitney U test for unpaired data, or chi Square test were used to estimate differences between groups. Differences were significant at *p* < 0.05. **dcSSc**, diffuse cutaneous SSc; **lcSSc**, limited cutaneous SSc; **DLCO% pred**, diffusion capacity of the lung for carbon monoxide as percentage of predicted; **FVC% pred**, forced vital capacity as percentage of predicted; **ILD**, interstitial lung disease; **mRSS**, modified Rodnan skin score; **LVEF**, Left ventricular ejection fraction; **CK**, creatine kinase; **E/A**, ratio of peak early and late transmitral flow velocities; **sPAP**, pulmonary artery systolic pressure; **CRP**, C-Reactive Protein; **BBDX incomplete**, incomplete block of the right branch; **ECG**, electrocardiogram. Statistically significant *p* values are in bold.

**Table 3 ijerph-19-10303-t003:** Quality of Life measurements.

Sections	All Patients (*n* = 34)	lcSSc (*n* = 25)	dcSSc *(n* = 9)	*p* Value *
**Phys_Funct**	76.0 ± 22.2	78.6 ± 23.4	69.4 ± 18.4	>0.05
**Role_limit_PH**	67.7 ± 40.4	72.7 ± 38.5	55.6 ± 44.7	>0.05
**Role_limit_EP**	69.9 ± 40.7	68.2 ± 40.5	74.1 ± 43.4	>0.05
**Energy/Fatigue**	55.2 ± 17.6	55.9 ± 17.1	53.3 ± 19.8	>0.05
**Emotional_WB**	62.3 ± 23.0	60.4 ± 20.7	67.1 ± 28.6	>0.05
**Social_funct**	68.9 ± 24.0	69.8 ± 23.8	66.7 ± 25.8	>0.05
**Bodily Pain**	65.2 ± 21.1	70.5 ± 20.7	52.2 ± 16.6	**0.026**
**General_health**	44.0 ± 17.9	45.0 ± 19.9	41.7 ± 12.5	>0.05
**Health Change**	53.2 ± 20.1	52.3 ± 20.3	55.6 ± 20.8	>0.05

Values are presented as mean ± SD. Depending on data distribution unpaired *t*-test or Mann–Whitney test was performed to determine significant differences. * *p* < 0.05 lcSSc vs. dcSSc. **Phys_Funct**, Physical functioning; **Role_limit_PH**, Role limitations due to physical health; **Role_limit_EP**, Role limitations due to emotional problems; **Emotional_WB**, Emotional well-being; **Social_funct**, Social functioning. Statistically significant *p* values are in bold.

**Table 4 ijerph-19-10303-t004:** Measurements of Individuals’ Habitual Physical Activity.

Sections		All Patients (*n* = 34)	*p* Value ^φ^	lcSSc (*n* = 25)	dcSSc (*n* = 9)	*p* Value *	*p* Value ^§^	*p* Value ^ψ^
**Work_index**	PRE	2.6 ± 0.6	**0.043**	2.5 ± 0.6	2.8 ± 0.7	>0.05	**0.048**	>0.05
	POST	2.3 ± 1.0		2.1 ± 1.1	2.7 ± 0.8	>0.05		
**Sport_index**	PRE	2.1 ± 0.7	>0.05	2.1 ± 0.8	1.9 ± 0.7	>0.05	>0.05	>0.05
	POST	1.9 ± 0.9		1.9 ± 1.0	2.0 ± 07	>0.05		
**Leisure_index**	PRE	2.8 ± 0.8	**0.009**	2.8 ± 0.9	2.8 ± 0.8	>0.05	**0.018**	>0.05
	POST	2.4 ± 1.1		2.3 ± 1.2	2.6 ± 07	>0.05		

Values are presented as mean ± SD. Depending on data distribution parametric or non-parametric test was performed to determine significant differences. * *p* < 0.05 lcSSc vs. dcSSc. ^§^ *p* < 0.05 PRE vs. POST within lcSSc group. ^ψ^ *p* < 0.05 PRE vs. POST within dcSSc group. ^φ^ *p* < 0.05 PRE vs. POST as whole. Statistically significant *p* values are in bold.

## Data Availability

The data presented in this study are available on request from the corresponding author. The data are not publicly available due to privacy or ethical restrictions.

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
