# Peer review of "The Preventive Role of Physical Activity in Systemic Sclerosis: A Cross-Sectional Study on the Correlation with Clinical Parameters and Disease Progression"

_ijerph, 2022, doi:10.3390/ijerph191610303_

Round 1
Reviewer 1 Report
Well written and interesting paper
Author Response
Manuscript ID: ijerph-1828188
Type of manuscript: Article
Title: The Preventive Role of Physical Activity in Systemic Sclerosis: A Cross-Sectional Study on the Correlation with Clinical Parameters and Disease Progression
Point-to-point answer to reviewers
We have really appreciated the revision performed by Reviewers because with their comments we believe to have significantly improved the quality of our manuscript. All changes were left tracked in the revised text with a note showing the reviewer's request.
REVIEWER 1
Well written and interesting paper
A0 - We thanks the Reviewer for the positive comments and evaluations.
Reviewer 2 Report
The manuscript is well-written and structured. However, major changes are needed.
Abstract
The abstract is well-written, as well as the manuscript itself. However, I suggest caution regarding the conclusion made by the authors. Since your data is based on self-reported measures, including retrospective aspects, and the design of the study did not allow a cause-and-effect, the authors must state the conclusion taken into consideration besides the mention related to physical load.
Introduction
This section is also well-structured, but I have some suggestions, as follows:
First paragraph
The sentences are long and most of them could be divided into two or more sentences. More concise sentences could help the reader. For example:
Systemic sclerosis (SSc) is a rare, systemic autoimmune disease characterized by skin fibrosis and vasculopathy [1]. Multiple systems (e.g., musculoskeletal, cardiovascular, pulmonary, and gastrointestinal) are involved which can result in a broad range of symptoms [2]. The prevalence of SSc ranges from 3.1 to 144.5 per 100,000 person-years, with a pooled 58 prevalence of 17.6 (95% CI, 15.1-20.5) per 100,000 person-years (I 2=100%), affecting mainly 59 women in the prime of their life [3,4]. SSc is associated with significant morbidity, including pain, disability, depression, and reduced quality of life [5-8]. Moreover, reduced physical activity (PA) and physical capacity in performing daily life activities have been also observed [9-11]. In fact, SSc may lead to muscle weakening [12] and impairment in oxygen transport and consumption [13], contributing significantly to reduced physical performance and poor health-related quality of life (HRQL) [14].
The same applies to the other paragraphs. I suggest the authors revised the Introduction section and shorten the sentences.
Regarding specifically the second and third paragraphs, I suggest merging them since both addresses the same topic.
Considering the previous studies about the theme, the authors mentioned limitations. However, it remains unclear whether your study could contribute to this field and also how the authors intend to overcome these limitations. Still, regarding the last paragraph, the authors could provide a more clear and more concise justification for the present study. I also suggest reporting the objectives as primary and secondary purposes.
Lastly, the authors did mention the subtypes of SSc in the section. However, one of your main goals includes them. Therefore, I strongly recommend that the authors provide some context regarding the subtypes, including the main differences between them.
Material and Methods
Although the authors have already mentioned the type of study in the Introduction section, I recommend adding the information on the methods. The data (mostly the questionnaires) was collected in only one of evaluation? I suggest including this information before the description of each measure.
Regarding the statistical analysis, I miss the sample size calculation and effect size.
Results
The SD should be placed near the mean. Please, revise page 4, lines 157 and 158.
In table 1, I suggest changing All patients to the Total sample, as well as adding the total number of participants.
After reading the results, I recommend adding a description for the clinical parameters in the Methods section, including the interesting variables and when, how, and by whom these measurements are taken.
In table 2, are the underlining and italics really necessary? Or just the bold?
Discussion
The discussion needs improvement.
I suggest the authors start this section by resuming the study's main purposes, followed by novelty comments and the main findings. Lastly, the authors could provide a brief statement about the practical implications, contribution to the field, and possible perspectives.
The first three paragraphs could be merged into one.
The authors failed to discuss the main findings or, at least, justify the results. As pointed out in the fourth paragraph, it seems that this study did not add to the knowledge about the relationship between PA and several disease aspects. In addition, the authors could at least add some possible explanations for the differences found, as well as the absence of differences for some variables.
It is equally essential to provide a proper discussion considering the proposed structure for the results, which can help your reader to understand the basis for your main findings. I recommend the authors revise this section to take that into account.
Conclusion and limitations
I recommend the authors provide the limitations separately from the conclusion. It is also interesting to include the potential strengths of the study and practical implications. Lastly, the conclusion must be concise and, as mentioned in the Abstract, the authors must be cautious and include the need for more studies with different designs investigating the role of PA for this population (i.e., as a preventive factor or helping to improve prognostic and/or delaying the progression of the disease, among others).
References
The references at the end of the manuscript present some errors, including a repetition of the number of each cited article. The authors must double-check the references.
Author Response
Manuscript ID: ijerph-1828188
Type of manuscript: Article
Title: The Preventive Role of Physical Activity in Systemic Sclerosis: A Cross-Sectional Study on the Correlation with Clinical Parameters and Disease Progression
Point-to-point answer to reviewers
We have really appreciated the revision performed by Reviewers because with their comments we believe to have significantly improved the quality of our manuscript. All changes were left tracked in the revised text with a note showing the reviewer's request.
REVIEWER 2
Q0 - The manuscript is well-written and structured. However, major changes are needed.
A0 - We have really appreciated this positive comment. Thank to Reviewer.
Abstract
Q1 - The abstract is well-written, as well as the manuscript itself. However, I suggest caution regarding the conclusion made by the authors. Since your data is based on self-reported measures, including retrospective aspects, and the design of the study did not allow a cause-and-effect, the authors must state the conclusion taken into consideration besides the mention related to physical load.
A1 - We thanks the Reviewer for this comment. As reported in the discussion section, we have already stressed the importance of high levels of physical load to determine improvement of numerous clinical parameters. Therefore, we have generalized in the abstract using the concept of physical activity. However, to avoid misleading concept, we revise the sentence in the abstract considering the conclusion taken into consideration.
Introduction
This section is also well-structured, but I have some suggestions, as follows:
First paragraph
Q2 - The sentences are long and most of them could be divided into two or more sentences. More concise sentences could help the reader. For example:
Systemic sclerosis (SSc) is a rare, systemic autoimmune disease characterized by skin fibrosis and vasculopathy [1]. Multiple systems (e.g., musculoskeletal, cardiovascular, pulmonary, and gastrointestinal) are involved which can result in a broad range of symptoms [2]. The prevalence of SSc ranges from 3.1 to 144.5 per 100,000 person-years, with a pooled 58 prevalence of 17.6 (95% CI, 15.1-20.5) per 100,000 person-years (I 2=100%), affecting mainly 59 women in the prime of their life [3,4]. SSc is associated with significant morbidity, including pain, disability, depression, and reduced quality of life [5-8]. Moreover, reduced physical activity (PA) and physical capacity in performing daily life activities have been also observed [9-11]. In fact, SSc may lead to muscle weakening [12] and impairment in oxygen transport and consumption [13], contributing significantly to reduced physical performance and poor health-related quality of life (HRQL) [14].
The same applies to the other paragraphs. I suggest the authors revised the Introduction section and shorten the sentences.
A2 - In agreement with the Reviewer, we revised all introduction section, making more concise and essential different sentences.
Q3 - Regarding specifically the second and third paragraphs, I suggest merging them since both addresses the same topic.
A3 - We substantially revised the second paragraphs being more general on the beneficial effects of regular exercise, while the third paragraph is focused on the specific disease condition.
Q4 - Considering the previous studies about the theme, the authors mentioned limitations. However, it remains unclear whether your study could contribute to this field and also how the authors intend to overcome these limitations. Still, regarding the last paragraph, the authors could provide a more clear and more concise justification for the present study. I also suggest reporting the objectives as primary and secondary purposes.
A4 - As we stated in our research article, here we presented an “exploratory study” to gather preliminary information that will help define problems and suggest hypotheses in anticipation of a longitudinal study capable to evaluate the effectiveness of a specific motor protocol on selected clinical parameters, indicators of the disease condition.
As suggested by Reviewer, we revised the last paragraph of the “Introduction”
Q5 - Lastly, the authors did mention the subtypes of SSc in the section. However, one of your main goals includes them. Therefore, I strongly recommend that the authors provide some context regarding the subtypes, including the main differences between them.
A5 - As suggested by Reviewer, we included a sentence clarifying the differences among SSc subtypes.
Material and Methods
Q6 - Although the authors have already mentioned the type of study in the Introduction section, I recommend adding the information on the methods. The data (mostly the questionnaires) was collected in only one of evaluation? I suggest including this information before the description of each measure.
A6 - We reported the type of study in the “Material and Methods” section, subsection 2.1 “Subjects”, last paragraph). Moreover, to better specify the data collection related to physical activity and quality of life assessment (SF-36), we added new sentences in the “Assessment of Habitual Physical Activity” section (first paragraph) and in the “Quality of life assessment (SF-36)” section (first paragraph), respectively.
Q7 - Regarding the statistical analysis, I miss the sample size calculation and effect size.
A7 - Since we envisioned this as an explorative study to determine if PA level changes were correlated/associated with demographic data (e.g., weight, height, gender, age, disease duration), diagnostic/clinical parameters (e.g., pulmonary hemodynamic/echocardiographic data, disease activity) related to disease progression, and quality of life measurements, no sample size calculation was performed. To clarify this aspect, we added a sentence in “Statistical analyses” section.
Results
Q8 - The SD should be placed near the mean. Please, revise page 4, lines 157 and 158.
A8 - We revised.
Q9 - In table 1, I suggest changing All patients to the Total sample, as well as adding the total number of participants.
A9 - In agreement with the Reviewer, we revised the table 1.
Q10 - After reading the results, I recommend adding a description for the clinical parameters in the Methods section, including the interesting variables and when, how, and by whom these measurements are taken.
A10 - We tanks the Reviewer for this comment because it allow us to better clarify this aspect. As we stated in “Material and Methods” section, all clinical parameters are investigated and recorded during periodic clinical visits by a rheumatologist. To avoid making the manuscript too long with widely known and repetitive methods in clinical practice, and similarly to other research articles focusing on the same topic (Azar et al., 2018; Filippetti et al., 2020; Pettersson et al., 2017), we have revised the sentence and summarily reported the bibliographic references relating to the methods of measuring clinical parameters.
References
Azar M, Rice DB, Kwakkenbos L, Carrier ME, Shrier I, Bartlett SJ, Hudson M, Mouthon L, Poiraudeau S, van den Ende CHM, Johnson SR, Rodriguez Reyna TS, Schouffoer AA, Welling J, Thombs BD; SPIN investigators. Exercise habits and factors associated with exercise in systemic sclerosis: a Scleroderma Patient-centered Intervention Network (SPIN) cohort study. Disabil Rehabil. 2018 Aug;40(17):1997-2003. doi: 10.1080/09638288.2017.1323023.
Filippetti M, Cazzoletti L, Zamboni F, Ferrari P, Caimmi C, Smania N, Tardivo S, Ferrari M. Effect of a tailored home-based exercise program in patients with systemic sclerosis: A randomized controlled trial. Scand J Med Sci Sports. 2020 Sep;30(9):1675-1684. doi: 10.1111/sms.13702. Epub 2020 May 11. PMID: 32350931; PMCID: PMC7496851.
Pettersson H, Åkerström A, Nordin A, Svenungsson E, Alexanderson H, Boström C. Self-reported physical capacity and activity in patients with systemic sclerosis and matched controls. Scand J Rheumatol. 2017 Nov;46(6):490-495. doi: 10.1080/03009742.2017.1281436.
Q11 - In table 2, are the underlining and italics really necessary? Or just the bold?
A11 - We removed the underlining and italics from the table 2.
Discussion
The discussion needs improvement.
Q12 -I suggest the authors start this section by resuming the study's main purposes, followed by novelty comments and the main findings. Lastly, the authors could provide a brief statement about the practical implications, contribution to the field, and possible perspectives.
A12 - As suggested by Reviewer, we revised this section, accordingly.
Q13 - The first three paragraphs could be merged into one.
A13 - As suggested by Reviewer, we revised this section, accordingly.
Q14 - The authors failed to discuss the main findings or, at least, justify the results. As pointed out in the fourth paragraph, it seems that this study did not add to the knowledge about the relationship between PA and several disease aspects. In addition, the authors could at least add some possible explanations for the differences found, as well as the absence of differences for some variables.
A14 - We thanks the Reviewer for this suggestion. We revised this section avoiding an excessive number of speculations within the manuscript based on not significant results (see Discussion section).
Q15 - It is equally essential to provide a proper discussion considering the proposed structure for the results, which can help your reader to understand the basis for your main findings. I recommend the authors revise this section to take that into account.
A15 - We thanks the Reviewer for this comment. As we reported in this version of our research article, the discussion follows the results structure highlighting mainly the relevant results. This allows us to avoid an excessive number of speculations within the manuscript (see Discussion section).
Conclusion and limitations
Q16 - I recommend the authors provide the limitations separately from the conclusion. It is also interesting to include the potential strengths of the study and practical implications. Lastly, the conclusion must be concise and, as mentioned in the Abstract, the authors must be cautious and include the need for more studies with different designs investigating the role of PA for this population (i.e., as a preventive factor or helping to improve prognostic and/or delaying the progression of the disease, among others).
A16 - As suggested by Reviewer, we revised this section and created two distinct sections, “Conclusion” and “Limitations”.
References
Q17 - The references at the end of the manuscript present some errors, including a repetition of the number of each cited article. The authors must double-check the references.
A17 - As suggested by Reviewer, we double--checked all references.
Reviewer 3 Report
Dear authors, congratulations on your work.
I have a few questions, namely:
- When do you assess PRE SSc diagnosis, what do you mean? How do you know that that person will develop SSc? I think this idea is not clearly described in your manuscript.
- You talk about disease duration. I suppose this is the time after de diagnostic. Individuals could be affected by the disease before diagnostic, or not?
- Page 9: when you say the name of 2 subfigures (e.g, A and B), the word "respectively" is missing
- Caption of figure 3: I suggest to represent the dashed and dotted lined as in the cation of figure 2.
- I felt that the discussion section could be more connected to the data presented in the results section. You can better support the discussion by referring to the objective results (which part of the results was on the basis of the sentences).
- I also felt that the discussion section could be improved (how do you take specific conclusions; possible reasons for example for the fact that literature does not present data about oxidative stress levels...). And also, the final hypothesis in section 4 must be deeply justified.
- At last, in the conclusion section, I do not understand the relevance of several studies has recruited fewer patients.
Author Response
Manuscript ID: ijerph-1828188
Type of manuscript: Article
Title: The Preventive Role of Physical Activity in Systemic Sclerosis: A Cross-Sectional Study on the Correlation with Clinical Parameters and Disease Progression
Point-to-point answer to reviewers
We have really appreciated the revision performed by Reviewers because with their comments we believe to have significantly improved the quality of our manuscript. All changes were left tracked in the revised text with a note showing the reviewer's request.
REVIEWER 3
Q1 - When do you assess PRE SSc diagnosis, what do you mean? How do you know that that person will develop SSc? I think this idea is not clearly described in your manuscript.
A1 - As we stated in the manuscript, this is a cross-sectional exploratory study where each patient recruited was asked to complete extensive questionnaires relative to individual’s perception of of their position in life in the context of the culture and value systems in which they live 112 and about their goals, expectations, standards, and concerns (also known as health-related 113 quality of life [HRQOL], through the Short Form Health Survey, SF-36) [41,42], as well as 114 to health-related physical activity performed before (PRE) and after (POST) the diagnosis 115 of disease through the modified Baecke Questionnaire for Older Adults (Material and Methods section, subsection 2.1 “Subjects”, last paragraph).
Moreover, to better specify this aspect, we added a new sentence in the “Assessment of Habitual Physical Activity” section (first paragraph).
Q2 - You talk about disease duration. I suppose this is the time after de diagnostic. Individuals could be affected by the disease before diagnostic, or not?
A2 - We have really appreciated this question because sometimes could be a long delay between the clinical onset and the diagnosis, especially for slow progressive SSc variants. In our study, we included only patients with the date of clinical onset close to diagnosis (< 1 month).
- Disease onset: the date of the first self-reported symptom (Raynaud phenomenon in the majority of patients).
- Disease diagnosis: the date when the patient fulfilled the European Scleroderma Trial and Re-102 search (EUSTAR) recommendations.
Q3 - Page 9: when you say the name of 2 subfigures (e. g, A and B), the word "respectively" is missing
A3 - We have revised this part of our paper accordingly
Q4 - Caption of figure 3: I suggest to represent the dashed and dotted lined as in the cation of figure 2.
A4 - We have really appreciated the Reviewer’s suggestion and understand the intention to uniform graphically the figures, however, we believe that the violin plot gives an immediate and easy interpretation of non-normal asymmetric variables distribution.
Q5 - I felt that the discussion section could be more connected to the data presented in the results section. You can better support the discussion by referring to the objective results (which part of the results was on the basis of the sentences).
A5 - As suggested from the Reviewer 3 (and Reviewer 2), we revised this section.
Q6 - I also felt that the discussion section could be improved (how do you take specific conclusions; possible reasons for example for the fact that literature does not present data about oxidative stress levels...). And also, the final hypothesis in section 4 must be deeply justified.
A6 - As suggested from the Reviewer, we revised this section.
Q7 - At last, in the conclusion section, I do not understand the relevance of several studies has recruited fewer patients.
A7 - We thanks the Reviewer for this comment. We revised this part of the manuscript.
Round 2
Reviewer 2 Report
The manuscript improved after revision. However, there are still minor changes needed.
Topic 2.1 could be named Study design and participants.
The authors should revise the formatting of the tables.
The Limitations section should be placed before the Conclusions.
Lastly, the Conclusion must be concise. The second paragraph could accompany the Limitations section, which, in this case, should be named Limitations and future perspectives.
Author Response
Manuscript ID: ijerph-1828188
Type of manuscript: Article
Title: The Preventive Role of Physical Activity in Systemic Sclerosis: A Cross-Sectional Study on the Correlation with Clinical Parameters and Disease Progression
2nd Round Point-to-point answer to reviewers 2
We really thanks the REVIEWER for his/her comments and suggestion.
We went through all suggestions and left tracked in the revised text with a note showing the reviewer's request.
REVIEWER 2
The manuscript improved after revision. However, there are still minor changes needed.
Q1 - Topic 2.1 could be named Study design and participants.
A1 - We revised the Topic 2.1
Q2 - The authors should revise the formatting of the tables.
A2 - As suggested by Reviewer, we revised all tables.
Q3 - The Limitations section should be placed before the Conclusions.
A3 - We revised the manuscript, accordingly.
Q4 - Lastly, the Conclusion must be concise. The second paragraph could accompany the Limitations section, which, in this case, should be named Limitations and future perspectives.
A4 - We revised the manuscript, accordingly.